# Once Read is Enough: Domain-specific Pretraining-free Language Models with Cluster-guided Sparse Experts for Long-tail Domain Knowledge

**Fang Dong**[1]   **Mengyi Chen**[1,*]   **Jixian Zhou**[1,*]   **Yubin Shi**[1]   **Yixuan Chen**[2]
**Mingzhi Dong**[3,1]   **Yujiang Wang**[2,†]   **Dongsheng Li**[4]   **Xiaochen Yang**[5]   **Rui Zhu**[6]
**Robert Dick**[7]   **Qin Lv**[8]   **Fan Yang**[9]   **Tun Lu**[1]   **Ning Gu**[1]   **Li Shang**[1,†]

[1]China and Shanghai Key Laboratory of Data Science, School of Computer Science, Fudan University
[2] Oxford Suzhou Centre for Advanced Research, Suzhou, China     [3] University of Bath
[4] Microsoft Research Asia, Shanghai, China
[5] School of Mathematics & Statistics, University of Glasgow
[6] Bayes Business School, City St George's, University of London
[7] Department of Electrical Engineering and Computer Science, University of Michigan
[8] Department of Computer Science, University of Colorado Boulder
[9] School of Microelectronics, Fudan University

## Abstract

Language models (LMs) only pretrained on a general and massive corpus usually cannot attain satisfying performance on domain-specific downstream tasks, and hence, applying domain-specific pretraining to LMs is a common and indispensable practice. However, domain-specific pretraining can be costly and time-consuming, hindering LMs' deployment in real-world applications. In this work, we consider the incapability to memorize domain-specific knowledge embedded in the general corpus with rare occurrences and "long-tail" distributions as the leading cause for pretrained LMs' inferior downstream performance. Analysis of Neural Tangent Kernels (NTKs) reveals that those long-tail data are commonly overlooked in the model's gradient updates and, consequently, are not effectively memorized, leading to poor domain-specific downstream performance. Based on the intuition that data with similar semantic meaning are closer in the embedding space, we devise a Cluster-guided Sparse Expert (CSE) layer to actively learn long-tail domain knowledge typically neglected in previous pretrained LMs. During pretraining, a CSE layer efficiently clusters domain knowledge together and assigns long-tail knowledge to designate extra experts. CSE is also a lightweight structure that only needs to be incorporated in several deep layers. With our training strategy, we found that during pretraining, data of long-tail knowledge gradually formulate isolated, "outlier" clusters in an LM's representation spaces, especially in deeper layers. Our experimental results show that only pretraining CSE-based LMs is enough to achieve superior performance than regularly pretrained-finetuned LMs on various downstream tasks, implying the prospects of domain-specific-pretraining-free language models.

---

*Co-first authors.

†Corresponding authors: <yujiang.wang@oscar.ox.ac.uk>, <lishang@fudan.edu.cn>.

# 1 Introduction

In natural language processing, it is a prevalent paradigm to pretrain language models (LMs) on a large-scale unlabeled corpus covering a plethora of knowledge, and those pretrained LMs have exhibited impressive performance in language tasks in the general domain [41]. When it comes to downstream tasks requiring specialized domain knowledge, e.g., legal search or medical question answering [23, 7], those models usually fail to expertise in such knowledge and cannot acquire desirable performance. As such, domain-specific pretraining on domain-specific datasets is deemed essential to fulfill pretrained LMs' potential in various downstream tasks [21, 14, 42, 36]. However, applying domain-specific pretraining on an LM could require domain expertise from humans, for instance, the involvement of a doctor for healthcare tasks [31], which can be costly and laborious. The associated catastrophic forgetting issue [26] could further complicate the domain-specific pretraining process.

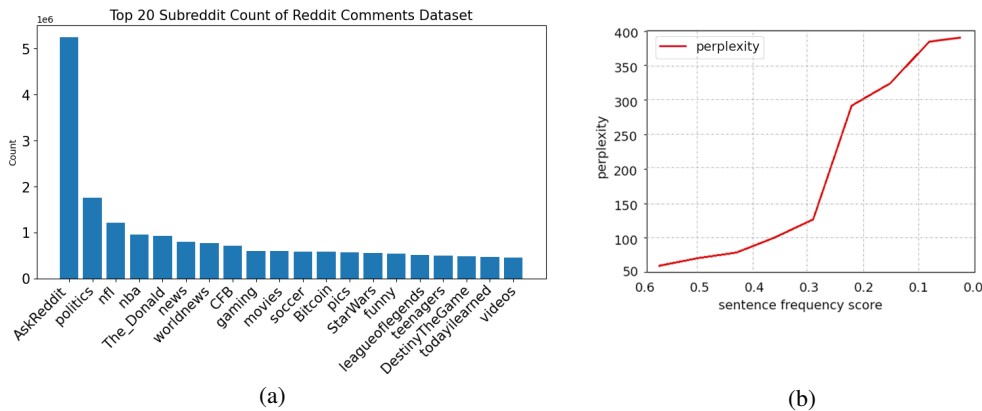

(a)                                    (b)

Figure 1: a) The top 20 subreddits with the highest amount of data in the Reddit Comments Dataset, where a typical long-tail distribution can be observed. b) Language Models struggle to memorize long-tail domain knowledge during pretraining. The less frequently a sentence appears in the training corpus, the higher its perplexity, indicating that it is not effectively memorized.

In this work, we re-visit the pretraining- domain-specific-pretraining paradigm and raise the following question: *is domain-specific pretraining indispensable to LMs*? Notably, the domain-specific knowledge necessary for various downstream tasks is usually embedded in the pretraining corpus of extensive information sources. Those pieces of domain-specific information may only appear a few times in the massive corpus, significantly less frequently than other ubiquitous and general knowledge, and there can be numerous pieces of such rare information, a distribution usually defined as "long-tail". In Fig 1(a), we plot the frequency of the top-20 subreddits count on Reddit Comments Dataset [2], and a typical long-tail distribution can be observed. Previous works have verified that LMs are not good learners of long-tail knowledge in the pretraining dataset with Question-Answering as the downstream task[20]. Our experiments, as shown in Figure 1(b), further illustrate that pretrained LMs do not adequately retain domain-specific knowledge in long-tail sequences which is evidenced by a surge in perplexity corresponding to decreased frequency score. This could result in inferior performance on downstream tasks. Domain-specific pretraining improves LMs' domain performance by providing a second lesson, which could be avoided if the first (pretraining) is appropriately delivered.

To unveil the hidden mechanisms under LM's incapability to learn long-tail domain-specific knowledge, we investigate the behaviors of a GPT on the Wikipedia dataset. We examine LMs' learning capabilities on long-tail data by analyzing the Neural Tangent Kernels (NTKs) of long-tail data and all data. Recent research [5, 40] has indicated that the updating of deep networks can be governed by the gradient direction corresponding to the principle eigenvector of an NTK matrix, which reflects the most common gradient-descending direction across the entire input space. Following those works, we consider an NTK's principle eigenvector (PE) gradient direction as a primary indicator of an LM's overall gradient-updating direction over a data space. Our analysis has revealed that the PE gradient direction of long-tail data, indicating the gradient-descending direction from long-tail knowledge, is

generally diverged from that of overall data, which rules the overall updating of network parameters. The observation that long-tail data cannot substantially impact LMs' parametric updates under regular pretraining settings explains pretrained LMs' incompetence on domain-specific knowledge of rare occurrences, necessitating an effective solution.

To this end, we propose the Cluster-guided Sparse Expert (CSE) layer, an effective, efficient, and easy-to-implement approach to improve LMs' long-tail knowledge awareness. In a CSE layer, with intuition such that data with similar semantic meaning are closer in the embedding space, we perform efficient clustering on the embeddings to group data from different domains, and additional experts will be assigned to explicitly and appropriately memorize the information within those clusters. Models trained with CSE show pronounced cluster structure in the embedding space, where long-tail data forms small, outlier clusters. We empirically demonstrate that converting several deep layers into CSE ones can be enough to achieve satisfying results, such as the last two layers of GPT[28] or BERT[10], and the incurred computational costs are comparatively small and arguably acceptable. We have verified that pretrained CSE-based LMs have outperformed regularly pretrained domain-specific pretrained LMs on downstream tasks from various domains, which implies that domain-specific pretraining may not be essential if long-tail knowledge can be sufficiently learned.

Our contributions are summarized as follows:

- We have presented that datasets show a long-tail distribution, with domain-specific data in the long-tail, and revealed that long-tail data cannot substantially affect LMs' training, which is a leading cause of LMs' incompetence in learning rare, domain-specific knowledge.

- We have devised a Cluster-guided Sparse Expert (CSE) architecture to better pretrain LMs to memorize the long-tail domain knowledge. With such a training strategy, LMs can effectively capture long-tail domain data in the representation space as outlier clusters, thereby enhancing their ability to handle less frequent contexts efficiently.

- Promising performance on downstream tasks has verified the effectiveness of the proposed method, indicating that domain-specific pretraining may not be indispensable to LMs.

## 2 Analysis of Long-Tail Domain Data

In this section, we first elucidate the challenges associated with learning from long-tail data through gradient analysis. We then explore the embedding space using the Cluster-guided Sparse Expert (CSE) layer, which effectively captures the structural nuances of long-tail data. Furthermore, we examine the dynamics of these clustering structures, offering insights into how the learning processes of long-tail clusters adapt and evolve across various training stages and model layers.

### 2.1 Challenges in Learning Long-Tail Domain Data

This subsection explores the significant challenges posed by long-tail domain data within language models (LMs). The primary issue stems from the divergence in gradient directions between long-tail data and the general gradient-updating trajectory of these models, which critically hampers effective learning.

#### 2.1.1 Preliminaries and Definitions

Informed by seminal works [12, 19], we utilize Neural Tangent Kernels (NTKs) to scrutinize the gradient behavior of neural networks under a gradient descent training regime. The NTK represented as $\Theta(\mathcal{X}, \mathcal{X})$, is defined as the outer product of the gradients of network outputs relative to its parameters $\Theta(\mathcal{X}, \mathcal{X}) = J_{\boldsymbol{\theta}}(\mathcal{X})J_{\boldsymbol{\theta}}(\mathcal{X})^{\top}$, where $J_{\boldsymbol{\theta}} = \nabla_{\boldsymbol{\theta}} f(\mathcal{X}; \boldsymbol{\theta})$ denotes the Jacobian matrix of the function $f$ at the data points $\mathcal{X}$.

To determine the predominant gradient-descending direction across the input space, which is influenced by the gradient direction associated with the principal eigenvector of the NTK matrix, we first perform an eigenvalue decomposition of the NTK matrix. Recognized as a positive semi-definite real symmetric matrix, the NTK decomposes into $\Theta = \mathbf{U}\boldsymbol{\Lambda}\mathbf{U}^{\top} = \sum_{i=1}^{n} \lambda_i \mathbf{u}_i \mathbf{u}_i^{\top}$. Here, $n$ represents the total number of training instances. The principal eigenvector $\mathbf{u}_{max}$ is identified as the vector corresponding to the maximum eigenvalue. Then the primary gradient direction for a given input set $\mathcal{X}$ is $\mathbf{g}_{\boldsymbol{\theta}}(\mathcal{X}) = \mathbf{u}_{max} J_{\boldsymbol{\theta}}(\mathcal{X})$. Building upon the above preliminaries, we introduce the metric of

Gradient Consistency (GC) to evaluate the alignment between gradient directions for specific data subsets and the overall dataset.

**Definition 1** *(Gradient Consistency (GC)).* Let $\mathcal{X}'$ *be a specific subset of the training set* $\mathcal{X}$. *The gradient consistency of* $\mathcal{X}'$ *is evaluated by computing the cosine similarity between the most prevalent gradient direction of* $\mathcal{X}'$ *and that of the entire dataset* $\mathcal{X}$:

$$GC_\theta(X') = \frac{\mathbf{g}_\theta(\mathcal{X}) \cdot \mathbf{g}_\theta(\mathcal{X}')}{\|\mathbf{g}_\theta(\mathcal{X})\|\|\mathbf{g}_\theta(\mathcal{X}')\|}. \tag{1}$$

A higher GC value indicates that the model's optimization updates are well-aligned with the needs of the specific subset $X'$, suggesting focused and effective learning of this data. Conversely, a lower value indicates suboptimal learning of these data, pointing to potential areas for improvement in model training strategies.

### 2.1.2 Gradient Consistency (GC) Analysis

We assess the sentences from Wikipedia on a standard GPT model using a sentence frequency score to gauge how frequently each sentence appears in the corpus. This score is calculated by averaging the frequency of its constituent tokens. Figure 2(a) displays the relationship between GC and sentence frequency score. Additionally, the figure includes a histogram that details how many percentages of sentences across the whole dataset fall into each frequency bin.

There is a significant correlation between gradient consistency and the frequency with which sentences appear in the corpus. Notably, for sentences less frequently encountered in the dataset, the model demonstrates substantial ineffectiveness in learning. As demonstrated, the GC value sharply declines from 0.8 to 0.4 as the sentence frequency score decreases from 0.3 to 0.2. Furthermore, the GC value continues to diminish as the sentence frequency score decreases further, indicating that the model's gradient descent direction struggles to align with the requirements of these rare sentences.

Our analysis indicates that the optimization requirements for long-tail sentences are significantly overlooked during standard pretraining, largely due to gradient conflicts between long-tail and common data. Research [38, 30] has demonstrated that these conflicts lead to suboptimal learning outcomes for the affected data. In typical pretraining, the gradient descent direction is dominated by common data, which prevents the model from effectively capturing the unique characteristics of long-tail domain data. This oversight significantly impairs the performance of LMs in learning domain-specific knowledge, particularly when dealing with rare occurrences, highlighting the need for a more effective solution.

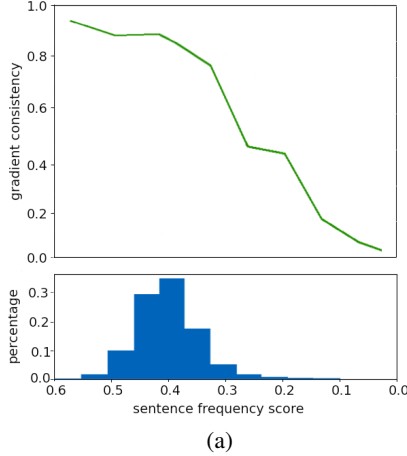
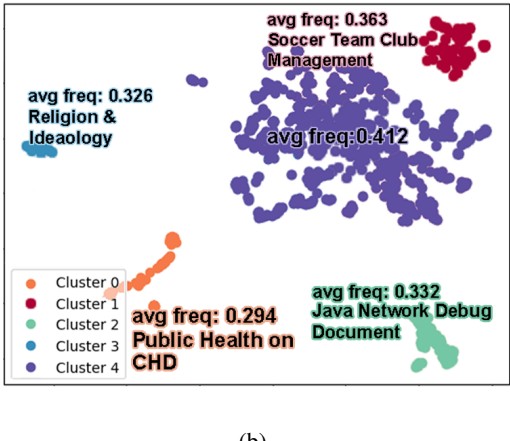

(a)          (b)

Figure 2:   a) The correlation between sentence frequency score and gradient consistency in the baseline model. A histogram is also included showing how many percentage of sentences across the whole dataset fall into each frequency bin. For further analysis using 2/3-gram averages, please refer to the appendix D.1. b) A sampled embedding space containing 4 long-tail clusters, taken from our CSE layers. For more information on the detailed cluster contents, please refer to the appendix D.2.

## 2.2 Embedding Space Analysis With Cluster-guided Sparse Expert (CSE) layer

Prior research[2] has shown that extensive domain-specific data reside within the long-tail distribution of a general pretraining corpus, as illustrated in Figure 1(a). These data, often semantically similar, are likely to cluster closely within the embedding space, facilitating potential aggregation for dedicated learning. However, our analysis in Section 2.1 underscores significant challenges in learning from long-tail data. Specifically, the model's gradient updates frequently fail to align with the optimization needs of these data, leading to their under-representation in the embedding space. Such misalignment obscures the inherent group structures that these domain data form based on their semantic similarities, thereby impeding dedicated learning efforts.

To address the issues outlined above and to facilitate a more effective examination of long-tail domain data in the embedding space, we propose the Cluster-guided Sparse Expert (CSE) layer. This layer groups proximate long-tail data points into clusters and directs them to specialized experts for dedicated learning. As demonstrated in Figure 3(a), the GC value of long-tail data in the baseline model initially increases at the beginning of the training stage but rapidly declines thereafter, indicating that the model's inability to capture the learning dynamics of long-tail data begins early in the training process. Our CSE layer capitalizes on the clustering structure at the point where the GC value peaks, subsequently taking effect to channel domain-specific clusters into dedicated learning pathways. Further details about this approach are provided in Section 3.

The clustering results from the CSE-based LM, shown in Figure 2(b), reveal four smaller clusters alongside a predominant one. Detailed analysis shows high domain coherence within the smaller clusters, each comprising sentences closely related to specific domains. The average sentence frequency score of these domain clusters falls into the long tail of the sentence frequency distribution, as shown in Figure 2(a). In contrast, the predominant cluster, colored in purple, contains a diverse mix of more common data and exhibits a higher average sentence frequency compared to the smaller clusters. Further analysis of sentences with frequency scores below 0.2 shows their random distribution across clusters, suggesting these extremely infrequent sentences may serve as noise in the learning process.

This analysis demonstrates that our proposed CSE-based architecture effectively groups long-tail data from the same domains for dedicated learning, fostering a domain-specific clustering structure within the embedding space. The long-tail domain clusters, distinct from clusters containing common data, show a higher degree of compactness and are clearly separated, highlighting the unique features embodied by these clusters.

## 2.3 Dynamic of Long-Tail Domain Clusters

In this subsection, we explore the learning dynamics of long-tail domain data by tracking how clusters evolve across different training stages and model layers. We utilize DBSCAN clustering[29] to determine the number of clusters.

**Long-tail clusters can be seen early in the training stage.** As shown in Figure 3(b) and Figure 3(c), the number of clusters quickly peaks early in the training stage, along with a peak in the ratio of cluster distances to cluster radii. This indicates that our CSE-based architecture effectively promotes the formation of a clustering structure early on.

The swift emergence of these clusters signifies substantial model adaptation to global features at the start of training, allowing for effective differentiation between clusters. As training progresses, inter-cluster distances gradually decrease, suggesting a stabilization in the learning dynamics and a potential shift in focus toward refining intra-cluster nuances.

**Long-tail clusters become more pronounced with increasing network depth.** Figures 3(b) and 3(c) demonstrate that the number of clusters is consistently higher in the deeper layers compared to the lower layers, with the ratio of cluster distances to cluster radii escalating significantly in the last two layers and reaching its maximum in the final layer. This pattern indicates that clusters become increasingly distinct and better separated as they progress through the network's layers. The enhanced separation of clusters in deeper layers can be attributed to the hierarchical feature extraction inherent in deep neural networks. As data moves through successive layers, the network abstracts and compiles more complex features, transitioning from general to more specific attributes. This

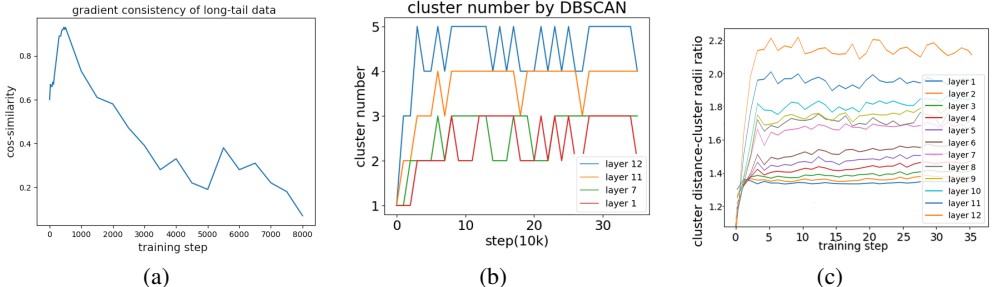

(a)  (b)  (c)

Figure 3: a) Evolution of the Gradient Consistency (GC) of long-tail data in the baseline model over the first 8000 training steps. GC scores beyond this range are omitted, as they consistently remain below 0.2. For details on the method used to select long-tail data, please refer to the appendix D.3. b) Evolution of number of clusters over training steps. c) The ratio of cluster distances to cluster radii over training steps, providing a measure of cluster structure clarity independent of norm values.

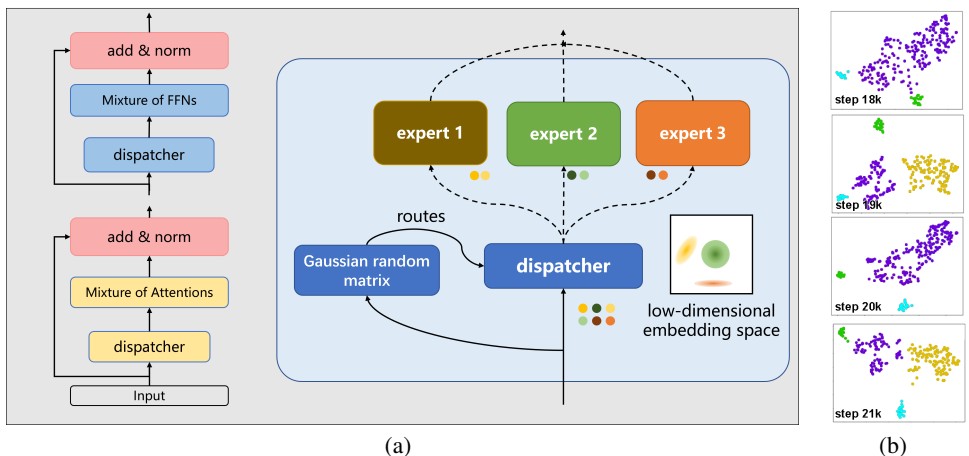

(a)  (b)

Figure 4: a) Overview of the Cluster-guided Sparse Expert (CSE) layer. b) The cluster number fluctuation is mainly caused by the big common cluster. These four figures arranged sequentially from top to bottom, were sampled at every 10,000 steps throughout the process from the FFN of the 10-th layer in a GPT model.

hierarchical processing allows the final layers to capture and enhance subtle distinctions between different data groups, leading to more defined and isolated clusters. This process not only underscores the capability of deep layers to refine and emphasize key features but also illustrates the network's efficiency in encoding progressively finer-grained information as layer depth increases.

## 3   Cluster-guided Sparse Expert (CSE)

To avoid the troublesome and costly domain-specific pretraining, we design a novel strategy, named **C**lsuter-guided **S**parse **E**xpert (CSE), to help the model capture the long-tail domain knowledge during pretraining. Since long-tail domain data show poor gradient consistency with overall data, we employ a sparse expert architecture within the Transformer model to assign data to different parameters, thereby avoiding the gradient conflict in each parameter group. This strategy can be applied to either attention or FFN. To dispatch data, with a straightforward and generally accepted intuition such that data with similar semantic meaning are closer in the embedding space, we design a very simple, efficient but effective online clustering algorithm operating concurrently with the language model pretraining, separate embeddings into different clusters, and use the outcome of this algorithm to instruct the dispatching of embeddings. The proposed algorithm is outlined in Algorithm 1.

**Dimension Reduction** In high-dimensional vector clustering, computational efficiency poses a significant challenge due to the $O(d^2)$ complexity of computing vector distance where $d$ denotes the dimensionality. So, we employ the same way of dimension reduction as is discussed in Section 2 before applying clustering on the embeddings, using a Gaussian random initialized matrix to project embeddings to a low-dimensional space[22]. This process, grounded in the Johnson-Lindenstrauss Lemma, effectively preserves the pairwise distances between embeddings while reducing their dimensionality, thereby enhancing the efficiency of our clustering algorithm.

**Initialization** We commence by training a baseline dense model devoid of any expert structure. Our findings in Section 2 illuminate an initial rise in gradient consistency between long-tail domain data and the general dataset at the onset of training, subsequently followed by a downturn. Consequently, we adopt a warm-up stage, letting the model learn the common features of long-tail and non-long-tail data. In our experiments, this process typically accounts for no more than 1% of the overall training. We then sample $N$ instances from the dataset and use its clustering result to initialize the cluster structure. We utilized DBSCAN

---

**Algorithm 1** Cluster-guided Sparse Expert
___
**Require:** $w$: Warm-up step count
**Require:** $N$: Initialization data count
**Require:** $M$: Gaussian random matrix $\in \mathbb{R}^{d \times d'}$ for reducing dimension
**Require:** $S$: Incoming embedding stream
**Require:** $\alpha$: center update factor
  1: Wait $w$ steps till the warm-up end.
  2: Sample $N$ data and run a clustering algorithm. Initialize cluster structure with the outcome by recording the cluster center $c_i$ and radius $r_i$ for each cluster.
  3: **for** $v$ in $S$ **do**
  4:    $v' = Mv$
  5:    $i = \arg\min_{j=1}^{C} ||v' - c_j||/r_j$
  6:    Dispatch $v$ to parameter group $i$
  7:    $c_i = \alpha c_i + (1 - \alpha)v'$
  8: **end for**

---

[29] in our experiments, a clustering algorithm that does not explicitly require the number of clusters. For every identified cluster, we document its centroid and define its radius as the average distance of all constituent data points from this central point

After this warm-up period, we fix the number of clusters and copy the module into cluster number copies. The module selection is introduced in the next paragraph. In our experiments, we noticed that the variations in the number of clusters were primarily driven by the splitting and merging of larger clusters, as illustrated in Figure 4(b); the smaller, long-tail clusters, however, remained largely unchanged. Consequently, adopting the initial clustering configuration directly, without further adjustments during training, was found to have no detrimental effect on model performance or the distribution of data handling. This approach capitalizes on the stability of the long-tail clusters and the dynamics of the larger ones, ensuring efficient data processing without compromising accuracy.

**Select Layer** Our motivation for performing clustering is rooted in the premise that semantically similar data tends to be closer. However, it is important to note that models learn the semantics of data progressively through layers; as we delve deeper into the model layers, the semantic information becomes increasingly rich, which may in turn amplify distinctions between data points. To quantify this variation, we apply our strategy only on layers with larger inter-cluster distances. Since the last 2 layers show a significant increase in inter-cluster distance, we apply our strategy in the last 2 layers, which is also the empirical best practice observed in existing moe-related works.[11, 25]

**Dispatch Embeddings** For each coming embedding, we decide the index of the expert it is dispatched to with $i = \arg\min_{j=1}^{n} ||v' - c_j||/r_j$, where $c_j$ denotes the center of cluster $j$, and $r_j$ denotes the radius of cluster $j$. Note that the $v'$ here is the sequence embedding rather than a token embedding and is defined as the mean of all token embeddings in the sequence[17], and the dispatching also happens on the sequence level. The rationale for defining and analyzing long-tail at the sequence level stems from the fact that when we discuss long-tails, we are essentially referring to semantics rather than individual words or tokens. Such semantics can only be observed within the context provided by sequences as a whole.

**Update cluster center** The model's parameter space undergoes gradual updates throughout training, causing a slow drift in the embedding space as the parameters evolve. To tackle this, we incorporate a dynamic mechanism to update the cluster centers concurrently with the assignment of clusters. For

a given cluster $mc_i$, let its center at time $t$ be denoted as $c_i^t$. When a new embedding $v$ arrives and is assigned to $mc_i$, we update $c_i^t$ with: $c_i^{t+1} = \alpha \cdot c_i^t + (1 - \alpha) \cdot v'$, where, $\alpha \in [0, 1]$ is a center update factor that determines the influence of the new embedding $v'$ on the existing center $c_i^t$. This adaptive updating scheme ensures that cluster centers remain representative of the current state of the embedding space, even as it evolves through the training process.

## 4 Related Works

**Long-Tail** Prior research addressing the issue of long-tail learning has predominantly been conducted within the domain of computer vision. The objective is to accurately recognize and classify rare or infrequently occurring classes in a given dataset together with frequently occurring classes [44]. There are several approaches to address the problem, including re-weighting [8], logit adjustment [4, 45], robust distributional matching [18, 35], and knowledge transfer [39, 34]. [37] declare that as the number of samples increases, the diminishing phenomenon suggests that there is a decreasing marginal benefit for a model to extract additional information from the data due to the presence of information overlap. Research in natural language processing has identified significant limitations in language models' capacity to learn long-tail knowledge [27, 3]. Furthermore, [46] suggests that attempting to address this issue during the domain-specific pretraining stage is often too late.

**Domain-Specific Pretraining** Domain-specific pretraining, also known as domain-specific pretraining, is highly advantageous to assist language models in requiring specialized domain knowledge. In one approach, contextualized embeddings are adapted to text from the target domain using masked language modeling, as detailed by Han and Eisenstein [16]. The concept of multi-phase pretraining involves secondary-stage unsupervised pretraining, exemplified by broad-coverage domain-specific BERT variants like BioBERT [24]. Research by Gururangan et al. [15] extends this by proposing domain-adaptive pretraining (DAPT) from a broader corpus and task-specific pretraining (TAPT) which uses unlabeled data increasingly aligned with the task distribution. These studies underscore the importance of domain-relevant data for pretraining in both high and low-resource scenarios [16, 15].

## 5 Experiments

This section presents the experimental results of our model and other methods. In the experiments, our model only undergoes a pretrained phase, reading domain-specific data once. Other methods are pretrained on the same dataset and then continue-pretrained on domain-specific datasets. Subsequently, all models are used as embedding models with all parameters frozen to generate embeddings for downstream tasks.

**Dataset and Evaluation** We employ Wikipedia [13] as our pretraining dataset, which is also widely accepted in other works [24, 10]. We adopt some legal and medical domain-specific downstream tasks to show the effectiveness of our model. To ensure that the pretraining data do contain domain knowledge required by the downstream tasks, we mixed a relatively small amount (less than 8%) of legal-domain-specific data [1] and medical-domain-specific data [9] into the pretraining data to simulate a long-tail distribution. The datasets selected are listed in Table 4 in Appendix A. Concurrently, we report the test perplexity of each model after the pretraining phase, serving as evidence of model convergence. Task performances are reported by accuracy. Although we use the method of mixing small-scale domain-specific datasets into pretraining data to simulate the long-tail distribution in those huge corpora, we cannot fully simulate the extremely rich pretraining data used on LLMs due to the limited training resources.

**Baselines** Since our strategy is not restricted to a specific model structure, we adopt both BERT [10] and GPT [28] as the base models and compare all the strategies on these base models respectively. We also compare with a Switch-MoE [11] version of them to show the effectiveness of our routing strategy. More Detailed implementation setting is listed in Appendix A.

### 5.1 Main Result

Table 9 and Table 2 show the performance of all models/strategies under our experiment setting with a trainable linear classifier for downstream tasks. **\*/med** means a model continue-pretrained

on medical-domain-specific data, and **\*/legal** means a model continue-pretrained on legal-domain-specific data. We tested Cluster-guided Sparse Expert on Attention and FFN respectively, denoted as **MoA** and **MoF**.

Table 1: Results of strategies applied on BERT

| Models | Pretrain ppl | Overruling | Casehold | GAD | EUADR | SST2 | average |
|---|---|---|---|---|---|---|---|
| BERT/med | 37.00 | 86.67 | 50.51 | 67.09 | 84.23 | 66.86 | 71.07 ± 0.22 |
| BERT/legal | 37.00 | 86.67 | 50.93 | 66.83 | 84.79 | 65.14 | 70.87 ± 0.23 |
| MoE/med | 31.00 | 85.00 | 50.49 | 64.52 | 83.10 | 64.79 | 69.58 ± 0.20 |
| MoE/legal | 31.00 | 85.83 | 50.30 | 64.32 | 84.79 | 63.88 | 69.82 ± 0.19 |
| Ours/MoA | 28.25 | 86.62 | **50.94** | **72.90** | 90.09 | 66.60 | 73.43 ± 0.18 |
| Ours/MoF | 34.64 | **89.10** | 50.82 | 71.65 | **91.23** | 67.98 | **74.16 ± 0.20** |

BERT/med exhibited a severe forgetting issue and details will be discussed in the Appendix A.

Table 2: Results of strategies applied on GPT

| Models | Pretrain ppl | Overruling | Casehold | GAD | EUADR | SST2 | average |
|---|---|---|---|---|---|---|---|
| GPT/med | 55.59 | 88.33 | 49.82 | 71.56 | 84.23 | 73.90 | 73.57 ± 0.17 |
| GPT/legal | 55.59 | 89.17 | 50.58 | 71.69 | 81.69 | 74.50 | 73.53 ± 0.23 |
| MoE/med | 40.69 | 91.25 | 50.11 | 72.77 | 83.38 | 72.03 | 73.91 ± 0.12 |
| MoE/legal | 40.69 | 91.60 | 49.68 | 72.66 | 83.38 | 71.97 | 73.86 ± 0.23 |
| Ours/MoA | 42.99 | 91.68 | 50.70 | 71.75 | **85.91** | 74.61 | 74.93 ± 0.08 |
| Ours/MoF | 43.38 | **93.33** | **51.26** | **73.30** | 85.63 | **76.00** | **75.90 ± 0.19** |

Our method outperforms other models/strategies on almost all tasks, with an average improvement of around 3%, showing an ability to learn long-tail data from the pretraining dataset. Our method can be applied to either the Attention module or the FFN module, and both ways will yield a better result compared with the domain-specific pretrained baselines, showing potential for eliminating the need for domain-specific pretraining. While in certain scenarios, domain-specific pretraining remains indispensable due to the privacy concerns associated with proprietary data, we argue that when pretraining datasets encompass domains similar to the proprietary one, our approach can still facilitate an enhanced domain-specific pretraining performance. We also present the result of a larger scale model, in Table 3. We used larger models (330M GPT-style models trained with 20B tokens) and domain-specific tasks from the academic, environmental, and financial domains to demonstrate the generalization capability of our method. The results indicate that our method, even without fine-tuning, consistently outperforms baselines by an average of 3-5% in accuracy on domain-specific tasks while maintaining comparable performance on general tasks. It is also notable that domain-specific pretraining leads to overfitting and even catastrophic forgetting, resulting in a decrease in performance on tasks from non-related domains. More details are shown in Appendix A. Further experiments show that our approach does not cause a reduction in general knowledge acquisition, and more details are shown in Appendix C.

Table 3: Results of strategies applied on 330M GPT

| Domain | Task | GPT/tuned | MoE/tuned | CSE/ w/o tune |
|---|---|---|---|---|
| academic | chem-prot | 36.25 | 36.25 | 36.25 |
| academic | MAG | 63.22 | 64.91 | **65.47** |
| academic | rct-20k | 76.95 | 78.28 | **80.15** |
| environment | clim. det. | 78.94 | 79.90 | **80.26** |
| environment | clim. sent. | 66.81 | 68.31 | **69.98** |
| financial | FPB | 16.83 | 25.00 | **40.11** |

## 5.2 Analysis

**Expert analysis** We analyze our model's embedding space to determine if our method dispatches embeddings correctly. We sample data and perform a forward inference pass through the model, visualizing the dispatching path of our model. As is shown in Figure 5, our distribution strategy correctly and effectively dispatches data from different long-tail clusters to different experts. We further visualize the NTK in each expert of our model, and it can be observed that by dispatching long-tail data separately, the NTK in each expert becomes more consistent. Whereas in a baseline model, its NTK matrix shows a poor consistency of the batch data, since long-tail and non-long-tail data are not separated.

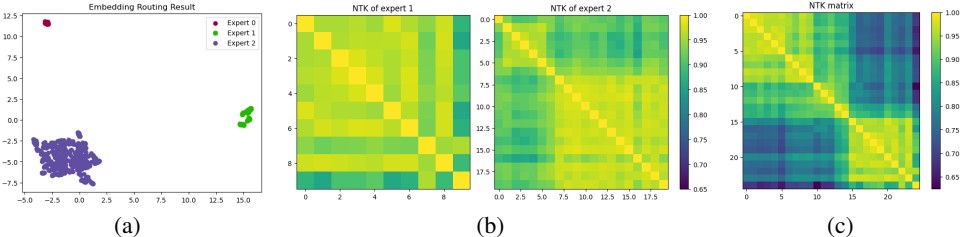

|(a)|(b)|(c)|

Figure 5: a) The embedding space and routing result of our model. b) The NTK in each expert in our model. c) The NTK in baseline. b) and c) are sampled from the FFN in the 10th layer.

For time overhead, the cluster initialization step is the additional computation introduced by our method throughout the training phase, relative to both the baseline and MoE. The time complexity of this operation will be $O(n \log n d')$ and is bounded by $O(n^2 d')$ for the worst case, where $n$ represents the number of sampled points. Notably, during subsequent inference stages, the quantum of parameters activated per iteration by our algorithm aligns identically with that of both MoE and the baseline. Since MoE and the baseline necessitate an additional fine-tuning phase, it becomes evident that the time cost of this phase far surpasses the $O(n^2 d')$ complexity of cluster initialization. To provide more straightforward data, the wall-clock time of training & domain-specific pretraining of 330M models for baseline GPT, MoE, and our CSE method and are 177 hours, 205 hours, and 160 hours respectively.

## 6   Conclusion

In this paper, we seek to elucidate why language models require domain-specific pretraining despite the presence of domain knowledge in their pretraining data. Our investigation uncovers that Sentences with lower frequency scores show diminished gradient consistency, resulting in increased test perplexity. This misalignment, particularly pronounced in low-frequency sentences, culminates in elevated test perplexity, suggesting a deficiency in effectively leveraging domain-specific information. To address this challenge, we introduce Cluster-guided Sparse Experts (CSE), grouping diverse long-tail domain data and dispatching them to different experts to enhance gradient consistency within each expert, thereby enabling the model to incorporate long-tail domain knowledge during pretraining. Experiments suggest that our approach has the potential to supplant the need for a dedicated domain-specific pretraining stage. Through this approach, long-tail domain instances promote the formation of small, outlier clusters in the representation space, exhibiting a characteristic signature across varying stages of training and architectural depths.

## Acknowledgements

This work was supported by the National Natural Science Foundation of China under Grant 62090025, and the Natural Science Foundation of Jiangsu Province (BK20240414).

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

# Appendix

# A   Experiments

Table 4 shows the datasets used in our experiments. Table 5 shows the hyperparameters used in our implementations. We use a machine with 8 NVIDIA GeForce RTX 3090 GPUs with 24GB GPU memory as our experiment platform. Pretraining costs about 24 hours.

Table 4: Datasets used for experiments

| Pretraining dataset | Description |
| --- | --- |
| Wikipedia ([13]) | Wikipedia dataset containing cleaned articles of all languages. The datasets are built from the Wikipedia dump with one split per language. Each example contains the content of one full Wikipedia article with cleaning to strip markdown and unwanted sections. |
| legal([1]) | In collaboration with Ravel Law, Harvard Law Library digitized over 40 million U.S. court decisions consisting of 6.7 million cases from the last 360 years into a dataset that is widely accessible to use. |
| PubMed([9]) | PubMed comprises more than 36 million citations for biomedical literature from MEDLINE, life science journals, and online books. |
| Finetuning task | Description |
| Overruling ([43]) | A law dataset corresponds to the task of determining when a sentence is overruling a prior decision. |
| Casehold([43]) | Case Holdings On Legal Decisions, comprising over 53,000+ multiple choice questions to identify the relevant holding of a cited case. |
| GAD([6]) | A relation extraction dataset, to decide if a gene is related to a specific disease. |
| EUADR([33]) | Another relation extraction dataset, to decide if a gene is related to a specific disease. |
| SST2([32]) | The Stanford Sentiment Treebank consists of sentences from movie reviews and human annotations of their sentiment. |

Table 5: Hyperparameters of Models

| Hyperparameters | BERT-based | GPT-based |
| --- | --- | --- |
| FFN modules | 4 | 6 |
| Attention modules | 4 | 6 |
| attention heads | 8 | 12 |
| our-strategy-based layers | 2 | 2 |
| transformer layers | 12 | 12 |
| Hidden dimension size | 768 | 768 |
| Droupt | 0.1 | 0.1 |
| Attention dropout | 0.1 | 0.1 |
| Sequence length | 128 | 256 |
| Batch size | 100 | 64 |
| Max steps | 36k | 300k |
| Learning rate decay | Cosine | Cosine |
| random seed used | 14, 24 | 22, 80 |

By monitoring the validation loss of the pretraining dataset(Figure 6), we show the Catastrophic Forgetting problem of the BERT model and its MOE method in the domain-specific finetuning phase. Despite our attempts at various combinations of generic data and domain-specific data during domain

finetuning, the best outcome among these still resulted in a decline in model performance on domains unrelated to its fine-tuning, indicating a limitation in the generalizability of the adapted model. As domain-specific finetuning proceeds, the validation loss of pretraining dataset has a significant rise and stays well above the convergence position of pretraining.

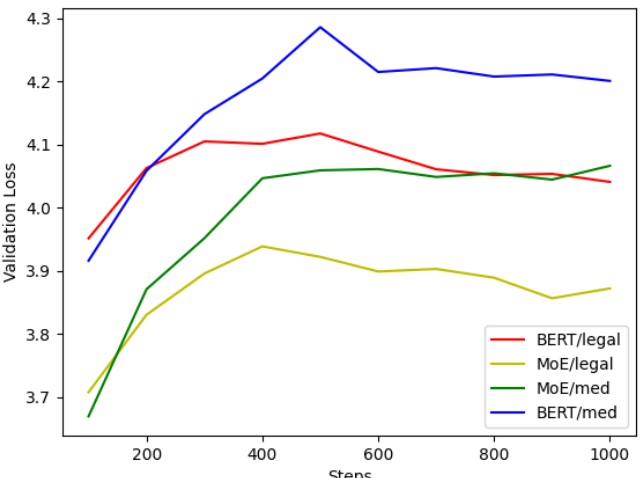

Figure 6: The validation loss of the pretraining dataset during the domain-specific finetuning phase.

Our training process strictly followed the basic framework of pretraining and domain-specific pretraining, and we didn't control the occurrence of forgetting, focusing more on the best performance. However as shown in the appendix, we monitored the forgetting phenomenon using the test loss on pretraining data. Specifically, BERT/med(further trained in medical), exhibited a more severe forgetting issue, with test perplexity on pretraining data increasing by 24.33, compared to a smaller increase of 8.41 for BERT/legal(further trained in legal). After correcting this imbalance by utilizing early-stop to select checkpoints where each model showed similar degrees of forgetting but not the best performance, the results came out that each domain-finetuned model outperformed on its respective domain tasks. This further underscores the challenges posed by the fine-tuning stage, affirming the value of our approach in not requiring domain-specific pretraining.

Table 6: Checkpoints selected with early-stop where two models show the same degree of forgetting.

| Models | Casehold | Overruling | GAD | EUADR |
|---|---|---|---|---|
| BERT/legal | **50.26** | **85.83** | 62.01 | 79.15 |
| BERT/med | 49.15 | 85.00 | **64.58** | **81.97** |

# B  Limitations Discussions

Although we use the method of mixing small-scale domain-specific datasets into pretraining data to simulate the long-tail distribution in those huge corpora, we cannot fully simulate the extremely rich pretraining data used on LLMs due to the limited training resources.

# C  Discussions

**Large Scale**  The maximum model scale presented in the experiment section in this paper is 330M GPT. So, we discuss what if our proposed CSE method is applied to a larger scale.

- **Model Size**: Larger models have greater learning capacities, which can enhance the GC for long-tail knowledge given a fixed training data size.
- **Window Size**: A larger window size allows the model to capture longer-range relationships within each training sequence, potentially highlighting more subtle distinctions and leading to a more long-tail distribution.
- **Training Data Size**: Larger data sizes and larger window sizes might result in a higher proportion of common data relative to long-tail clusters. The increased presence of common data and the potential subdivision of long-tail clusters into smaller groups could, paradoxically, reduce the GC for long-tail knowledge.

**Performance on Common Datasets**  Additionally, we have included results on general knowledge tasks to confirm that our dedicated long-tail learning method does not compromise general knowledge acquisition.

Table 7: Results of general tasks on BERT with the same small-scale setting in the paper

| Task | Domain | Freq. Score | Baseline(tuned) | MoE(tuned) | Ours(w/o tune) |
|------|--------|-------------|-----------------|------------|----------------|
| COLA | general | 0.389 | 69.22 | 69.32 | **69.42** |
| QNLI | general | 0.325 | 65.92 | 65.07 | **66.04** |
| MRPC | general | 0.343 | **72.06** | 70.83 | **72.06** |
| QQP | general | 0.380 | **70.64** | 69.85 | 70.18 |
| SST2 | general | 0.327 | 66.86 | 64.79 | **67.98** |
| average | general | - | 68.94(-0.20) | 67.97(-1.26) | **69.14** |

Table 8: Results of general tasks tested on GPT 330M trained with 20B tokens

| Task | Domain | Freq. Score | Baseline(tuned) | MoE(tuned) | Ours(w/o tune) |
|------|--------|-------------|-----------------|------------|----------------|
| COLA | general | 0.389 | 69.10 | 69.10 | **69.20** |
| QNLI | general | 0.325 | **60.17** | 60.06 | 59.72 |
| MRPC | general | 0.343 | 70.18 | 71.75 | **71.98** |
| QQP | general | 0.380 | 73.28 | 74.47 | **75.95** |
| SST2 | general | 0.327 | 74.50 | 72.03 | **76.00** |
| average | general | - | 69.45(-1.12) | 69.48(-1.09) | **70.57** |

**Experiments on A Pre-trained Model**  We also conducted experiments on a pre-trained 110M scale model, wherein all methods continue training from a single pre-trained checkpoint. The outcomes are presented in the following table. Results show that directly applying our method to a pre-trained model still yields superior performance compared to the baseline and MoE models.

Table 9: Results of strategies applied on pre-trained model

| Models | Overruling | Casehold | GAD | EUADR | SST2 | average |
|--------|-----------|----------|-----|-------|------|---------|
| BERT/med | 85.00 | 51.11 | 70.84 | 89.86 | **64.11** | 72.18 ± 0.21 |
| BERT/legal | 85.83 | **51.21** | 66.08 | 88.45 | 61.58 | 70.63 ± 0.17 |
| MoE/med | 85.83 | 49.91 | 72.18 | 90.42 | 63.42 | 72.35 ± 0.18 |
| MoE/legal | **86.67** | 50.83 | 69.44 | 89.86 | 62.16 | 71.88 ± 0.17 |
| Ours/MoA | **86.67** | 51.11 | 73.17 | **92.96** | 63.99 | **73.58 ± 0.19** |
| Ours/MoF | **86.67** | 50.77 | **73.25** | 91.83 | 63.65 | 73.23 ± 0.18 |

**Using both MoA and MoF**  We tried this architecture using both MoA and MoF and tested it in the same small-scale setting reported in the paper, and found that MoA+MoF yields improvements in the overall performance compared to MoA/MoF only.

Table 10: Updated results of strategies applied on BERT

| Models | Pretrain ppl | Overruling | Casehold | GAD | EUADR | SST2 | average |
|---|---|---|---|---|---|---|---|
| MoA | 28.25 | 86.62 | **50.94** | 72.90 | 90.09 | 66.60 | 73.43 |
| MoF | 34.64 | **89.10** | 50.82 | 71.65 | 91.23 | **67.98** | 74.16 |
| MoA+MoF | 28.25 | 87.50 | 50.83 | **79.87** | **93.80** | 67.09 | **75.82** |

# D  Analysis of Sentence Frequency Distribution and Cluster Details

## D.1  Sentence Frequency Distribution and Gradient Consistency

We analyze the sentence frequency distribution using 2/3-gram averages, comparing it to gradient consistency across different training stages. Figure 7(a) and Figure 7(b) illustrate the frequency distribution of sentences based on their 2-gram and 3-gram patterns, respectively. The results are consistent with the 1-gram analysis, confirming that despite its simplicity, the 1-gram method remains effective in capturing sentence frequency trends. The gradient consistency results align with these findings, reinforcing the robustness of this approach.

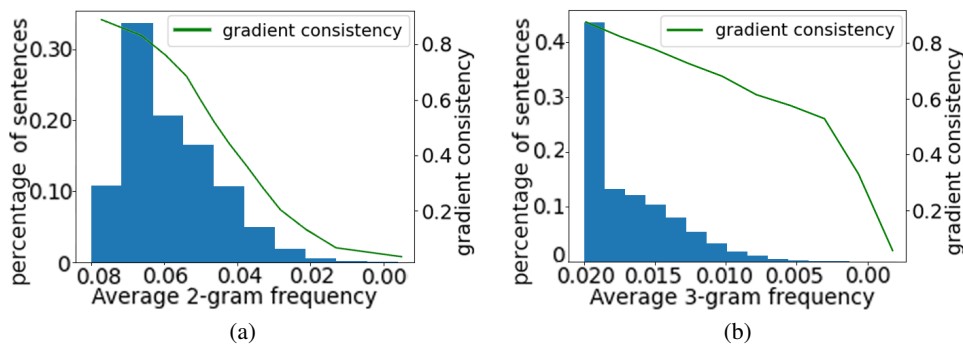

Figure 7:  a) 2-gram sentence frequency distribution. b) 3-gram sentence frequency distribution.

## D.2  Cluster Details and Sentence Content

Figure 8 shows the detailed content of sentences within each small cluster. For example, in the 'Java Network Debug' cluster, terms such as HTML, Javascript, and `<link>` frequently appear, indicating strong domain coherence. This example highlights how specific topics tend to cluster together, demonstrating the effectiveness of the clustering process.

Furthermore, we identified low-frequency sentences that are scattered across clusters, lacking clear semantic correlation. These sentences often include irregular content such as misprinted formulas or non-English text, which prevents them from forming coherent clusters. Figure 8 also illustrates the distribution and content of these low-frequency sentences, providing insight into their irregular placement across clusters.

## D.3  Methodology for Long-Tail Data Selection

We applied the Elbow method to determine a threshold of 9.37% on the curve of domain proportions, classifying any domain with a data proportion at or below this level as a long-tail domain. By plotting the curve of Wikipedia domains and their corresponding proportions, we identified the point (9.37%) where the slope changes significantly, marking the transition from the head of the distribution to the tail. Following this, we randomly selected sentences from these long-tail domains for further analysis.

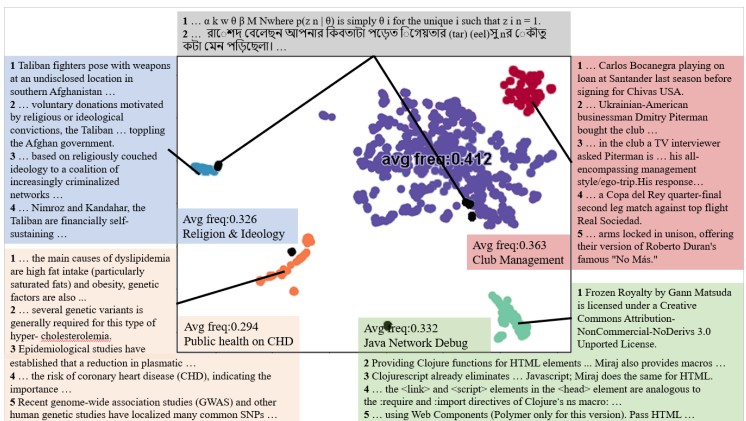

Figure 8: Detailed content of sentences within clusters, showing high domain coherence (e.g., 'Java Network Debug' cluster) and the distribution of low-frequency, irregular sentences.

