# OpenReview forum: "Once Read is Enough: Domain-specific Pretraining-free Language Models with Cluster-guided Sparse Experts for Long-tail Domain Knowledge"
_NeurIPS.cc/2024/Conference — NeurIPS 2024 poster_

### Official Review · Reviewer_P2qX · 2024-06-29

**Soundness:** 2
**Presentation:** 1
**Contribution:** 1
**Rating:** 4
**Confidence:** 3

**Summary:**

This paper first shows that infrequent sentences tend to have gradients dissimilar to the gradient of the majority data points. This paper thus proposes a variant of mixture-of-expert (MoE) module,  Cluster-guided Sparse Expert (CSE), where inputs are routed to the k-mean clusters they are mostly similar to. They experiment with BERT and GPT and show that their proposed CSE moduled outperform Switch-MoE.

**Strengths:**

1. Albeit the simplicity of the way they compute sentence frequency, they are still able to show that less frequent sentences have gradients/Jacobian less similar to the gradient of majority.
2. The proposed CSE is simple and does not incur much extra computational overhead.

**Weaknesses:**

**The Experiment Setup**

1. First, the setup is not clear. I am not sure if this work follows the pretrain-and-fine-tune paradigm for both BERT and GPT.
2. The more common use case of MoE is for language modeling, which is not covered by this work.
3. The baseline should include an MoE model pretrained with Wikipedia.

**Analyses**

1. The sentence frequency score is just the average of 1-gram frequency, which may be too simple.
2. Compared with Figure 2a, the consistency in Figure 3a, which I suppose is the result of using CSE, does not seem to be better. This contradicts their motivation for using CSE.
3. The increased cluster distance shown in Figure 3c may not be very meaningful, as the distance can be larger simply because the norm of the representations is larger.
2. Some arguments are not substantiated. For example:
    1. line 152:  “Detailed analysis shows high domain coherence within the smaller clusters each comprising sentences closely related to specific domains.” The detailed analysis is not specified.
    2. line 157: “Further analysis of sentences with frequency scores below 0.2 shows their random distribution across clusters, suggesting these extremely infrequent sentences may serve as noise in the learning process.” The meaning of “random distribution” is unclear, and the details about the analyses are also absent.
3. It is not that surprising that samples in the same cluster have similar gradients, because if $x \approx y$ then as long as $f$ is smooth in a certain way, we have $f’(x) \approx f’(y)$.

**Presentation/Writing**

1. Section 2.2 analyzes their CSE module, but the CSE module is formally introduced in Section 3, which makes the paper hard to follow.
2. What are line 1 to line 12 in Figure 3c?
3. The font size of the text in the figures is too small.
4. The authors should discuss why they choose the testing datasets and the domain of those datasets in the main text.
5. There are many unsubstantiated or unclear sentences. For example:
    1. line 128: “the optimization requirements for long-tail sentences are significantly overlooked under standard pretraining condition” not substantiated
    2. line 146:  the model’s inability to capture the learning dynamics of long-tail data begins early in the training process
    3. line 148: “Our CSE layer capitalizes on the clustering structure at the point where the GC value peaks” I can’t understand.

**Misc**

The title “Once Read is Enough” seems to be over-claiming.

The authors may want to read/cite this paper
- On the Benefits of Learning to Route in Mixture-of-Experts Models

**Questions:**

Do you add a classification head and fine-tune the whole model?

**Limitations:**

In Appendix B.

---

> ### Author Rebuttal · Authors · 2024-08-06
>
> We appreciate your thorough review of our paper. Our key contribution addresses the challenge of learning long-tail domain data during the pretraining stage, substantially reducing the need for an expensive, labor-intensive second domain-specific pretraining stage in practical applications.
> > Q1 ..setup is not clear
>
> Both BERT and GPT undergo both pretraining and domain-specific pretraining, while our method only pretraining. All models then serve as embedding generators with all params frozen for downstream tasks.
> > Q2 ..language modeling not covered
>
> Our BERT-based MoE uses MLM for pretraining, while the GPT-based MoE employs causal language modeling. All downstream tasks are NLP tasks. We are unclear about the specific meaning of 'language modeling' you mentioned and appreciate further clarification.
> > Q3 .. MoE pre-trained with Wikipedia.
>
> We compared MoE with 1) only pretraining 2) additional domain-specific pretraining and 3) our method. Results show that while domain-specific pretraining boosts baseline performance, our single-stage pretraining method surpasses both, efficiently learning long-tail domain.
>
> |Models|Overruling|Casehold|GAD|EUADR|SST2|avg|
> |-|-|-|-|-|-|-|
> |MoE/med|91.25|50.11|72.77|83.38|72.03|73.91|
> |MoE/legal|91.60|49.68|72.66|83.38|71.97|73.86|
> |MoE|91.33|49.10|71.70|83.10|72.28|73.50|
> |Ours/MoA|91.68|50.70|71.75|**85.91**|74.61|74.93|
> |Ours/MoF|**93.33**|**51.26**|**73.30**|85.63|**76.00**|**75.90**|
> > Q4 .. 1-gram frequency too simple.
>
> Global rebuttal Figure3 displays sentence frequency distribution using 2/3-gram averages, alongside gradient consistency. Results align with 1-gram analysis, confirming its effectiveness despite simplicity.
> > Q5 Compared with Figure2a, consistency in Figure3a..not..better..
>
> Both figures were obtained on a baseline model. We'll clarify this in the main text.
> > Q6 cluster distance not meaningful..norm..larger.
>
> In global rebuttal Figure2, we introduce a new metric—the ratio of cluster distances to cluster radii—to measure cluster structure clarity independent of norm values. This aligns with our observation that clusters become more distinct in deeper layers.
> > Q7 line 152:"..high domain coherence within the smaller clusters.." not specified.
>
> In global rebuttal Figure4, we detail the content of sentences within each small cluster. For example, 'Java Network Debug' cluster frequently includes HTML, Javascript, and <link>, demonstrating high domain coherence.
> > Q8 line 157:"..frequency scores below 0.2 shows random distribution.."..random distribution unclear.
>
> The term "random distribution" describes how extremely low-frequency sentences are irregularly scattered across clusters without clear semantic correlation. In Global rebuttal Figure4, we detailed the distribution and contents of these sentences, which often contains misprinted formulas or non-English.
> > Q9 not that surprising that samples in the same cluster have similar gradients.
>
> Our main contribution is that we are the first to apply gradient similarity to identify and tackle the challenges of learning long-tail data during pretraining, significantly reducing the need for subsequent domain-specific pretraining.
> > Q10 Section 2.2 analyzes their CSE module .. formally introduced in Section 3..hard to follow.
>
> Our goal was to ensure readers first understand the fundamental challenges of long-tail data learning, which sets the stage for appreciating the effectiveness of our method. We'll include a more detailed illustration of CSE in analysis section of revised version.
> > Q11 ..line 1 to line 12 in Figure3c?
>
> We'll correct "line" to "layer".
> > Q12 The font size..in the figures is too small.
>
> We'll increase the font size in figures.
> > Q13 ..discuss why..choose the testing datasets..domain of those datasets in the main text.
>
> Our research identified long-tail distribution as a prevalent phenomenon in many common knowledge datasets, posing a challenge for pretraining. We chose Wikipedia for pretraining because it mirrors this distribution pattern. We also chose specific domains within the long-tail distribution with publicly available training corpora and downstream tasks for further testing. A detailed explanation of dataset choices will be included in the main text in revised version.
> > Q14 ..line 128:"optimization requirements for long-tail..overlooked .." not substantiated.
>
> Existing researches [A, B] has shown that gradient conflicts lead to suboptimal learning outcomes for the affected data. In standard pretraining, since the gradient descent direction is dominated by common data, the differences in gradient direction between long-tail and common data result in the optimization of long-tail data being overlooked.
> > Q15 ..line 146: model's inability..long-tail..begins early..
>
> Gradient consistency indicates how much a model focuses on learning specific data. Figure3a in the paper reveals that the GC of long-tail data declines sharply at the beginning of the pretraining stage and remains low, suggesting that the model's inability to effectively learn from long-tail data starts early in the training process.
> > Q16 ..line 148:"CSE layer..at..GC value peaks"
>
> The peak GC value of a baseline model signals when it struggles to learn long-tail data. We then introduce CSE layer at this point to improve the model’s ability to learn long-tail data in later phases.
> > Q17 title..over-claiming
>
> We'll replace 'finetuning' with 'domain-specific pretraining' to narrow down the specific training stage we aim to reduce.
> > Q18 read/cite this paper ..
>
> We find this paper relevant and will cite it.
> > Q19 Do you add a classification head and fine-tune the whole model?
>
> For each classification task, we train a classification head with all other params frozen.
>
> **References**
>
> [A] T. Yu, et al. Gradient Surgery for Multi-Task Learning. NeurIPS 2020.
>
> [B] G. Shi, et al. RECON: REDUCING CONFLICTING GRADIENTS FROM THE ROOT FOR MULTI-TASK LEARNING. ICLR 2023.

---

> > ### Comment · Reviewer_P2qX · 2024-08-07
> >
> > Thanks for the clarifications.
> >
> > - Q2: One of the most common usages of pretrained LMs is to use them without any fine-tuning. People use them to solve tasks by prompting the LMs. Unfortunately, it is not discussed in this paper.
> > - Q5: For Figure 3(c), the authors should also specify what criteria/threshold they used for choosing the "long-tailed" data.
> > - Q19: "For each classification task, we train a classification head with all other params frozen." This is a uncommon setup now.
> >
> > In sum, I appreciate the authors' clarification and the provided extra experimental results. Taking this into account, I increased my score. However, at the same time, I would like to mention
> >
> > 1. I am not sure if the next revision can incorporate all the clarifications and the new results/metric well. In my opinion, the current writing quality is not ready for publication for a top conference such as NeuRIPs.
> > 2. The setup this paper studies, aka a 300M pretrained model as a feature extractor, is not mainstream, at least in academia.
> >
> > I would leave the decision to AC.

---

> ### Author Response · Authors · 2024-08-10
>
> Thank you for your comments.
>
> > DQ1 One of the most common usages of pretrained LMs is to use them without any fine-tuning. People use them to solve tasks by prompting the LMs. Unfortunately, it is not discussed in this paper.
>
> __DA1__ We share your view that eliminating fine-tuning is highly valuable. While prompting is a common approach, domain-specific fine-tuning is also widely adopted, particularly for downstream tasks that require specialized domain knowledge[J, K, L, M, N, O]. In these cases, models often struggle to achieve the desired performance without specific expertise in the relevant domain. Fine-tuning is therefore essential to unlock the full potential of pre-trained language models for various downstream tasks.
>
> Our work aims to minimize the need for cumbersome domain-specific fine-tuning, especially when models have already encountered relevant long-tail domain data during pretraining but have not effectively learned from it. In such cases, prompting alone may fall short in recovering lost knowledge without fine-tuning. Our method addresses this challenge by enhancing the model's ability to learn and retain long-tail domain knowledge during pretraining, thereby reducing the necessity for extensive domain-specific fine-tuning. Consequently, our approach provides a more robust solution, preserving and enriching critical domain knowledge, and offering a distinct advantage over purely prompting-based methods.
>
> > DQ2 For Figure 3(c), the authors should also specify what criteria/threshold they used for choosing the "long-tailed" data.
>
> __DA2__ Thank you for your question. We applied the Elbow method to determine a threshold of 9.37\% on the curve of domain proportions, classifying any domain with a data proportion at or below this level as a long-tail domain. By plotting the curve of Wikipedia domains and their corresponding proportions, we identified the point (9.37\%) where the slope changes significantly, marking the transition from the head of the distribution to the tail. Following this, we randomly selected sentences from these long-tail domains for further analysis.
>
> > DQ3 "For each classification task, we train a classification head with all other params frozen." This is a uncommon setup now.
>
> __DA3__ We share your perspective that there are other prevalent paradigms, but our approach is also widely adopted. The original BERT paper [I] mentions the feature-based fine-tuning approach, where "fixed features are extracted from the pre-trained model" and a "task-specific model architecture" is added and fine-tuned for specific tasks, such as a classification head for classification tasks. Similarly, the standard paradigm for training domain-specific models also follows this fine-tuning approach after a domain-specific pretraining stage. For instance, Section 4.1 in [A], Section 3 in [C], Section 3.3 in [G], Section 2.1 in [F], and other references [B, D, E, H] all explicitly state that they adhere to this paradigm.
>
> > DQ4 I am not sure if the next revision can incorporate all the clarifications and the new results/metric well. In my opinion, the current writing quality is not ready for publication for a top conference such as NeuRIPs.
>
> __DA4__ Thanks for your question. We've summarized review comments received below and have already completed them in the revised version. We will incorporate further refinements as new comments come in.
>
> |Category | Modification | Status |
> |-|-|-|
> |Experiments | Add large-scale experiment results | Completed |
> |Experiments | Include a pre-trained only baseline | Completed |
> |Metrics | Replace 'inter-cluster distance' with 'ratio of cluster distances to cluster radii' | Completed |
> |Revisions | Clarify that Figures 2a and 3a are obtained using baseline models | Completed |
> |Revisions | Provide rationale for dataset selection | Completed |
> |Revisions | Introduce CSE when GC value peaks and explain why | Completed |
> |Revisions | Detail Figure 2b with specific sentences in long-tail clusters and describe the random distribution of extremely low-frequency noisy data points | Completed |
> |Revisions | Elaborate on CSE method in the analysis section | Completed |
> |Revisions | Incorporate references and summarize research on gradient conflicts | Completed |
> |Typo/Figure Font | Correct typographical errors and increase figure font sizes | Completed |
>
> Continued on the next post

---

> > ### Author Response · Authors · 2024-08-10
> >
> > > DQ5 The setup this paper studies, aka a 300M pretrained model as a feature extractor, is not mainstream, at least in academia.
> >
> > __DA5__ We share your perspective that there are other prevalent paradigms, but our approach of using a pre-trained model as a feature extractor is also widely adopted. The original BERT paper [I] mentions the feature-based fine-tuning approach, where "fixed features are extracted from the pre-trained model" and a "task-specific model architecture" is added and fine-tuned for specific tasks, such as a classification head for classification tasks. Similarly, the standard paradigm for training domain-specific models also follows this fine-tuning approach after a domain-specific pretraining stage. For instance, Section 4.1 in [A], Section 3 in [C], Section 3.3 in [G], Section 2.1 in [F], and other references [B, D, E, H] all explicitly state that they adhere to this paradigm. Additionally, these papers also use models with millions of parameters (such as 300M) as their backbone.
> >
> >
> > __References__
> >
> > [A] Gururangan, Suchin, et al. "Don't stop pretraining: Adapt language models to domains and tasks." arXiv preprint arXiv:2004.10964 (2020).
> >
> > [B] Gu, Yu, et al. "Domain-specific language model   pretraining for biomedical natural language processing." ACM Transactions on Computing for Healthcare (HEALTH) 3.1 (2021): 1-23.
> >
> > [C] Webersinke, Nicolas, et al. "Climatebert: A pretrained language model for climate-related text." arXiv preprint arXiv:2110.12010 (2021).
> >
> > [D] Jørgensen, Rasmus Kær, et al. "mdapt: Multilingual domain adaptive pretraining in a single model." arXiv preprint arXiv:2109.06605 (2021).
> >
> > [E] Krieger, Jan-David, et al. "A domain-adaptive pre-training approach for language bias detection in news." Proceedings of the 22nd ACM/IEEE joint conference on digital libraries. 2022.
> >
> > [F] Bayer, Markus, et al. "Cysecbert: A domain-adapted language model for the cybersecurity domain." ACM Transactions on Privacy and Security 27.2 (2024): 1-20.
> >
> > [G] Lamproudis, Anastasios, Aron Henriksson, and Hercules Dalianis. "Vocabulary Modifications for Domain-adaptive Pretraining of Clinical Language Models." HEALTHINF. 2022.
> >
> > [H] Konlea, Leonard, and Fotis Jannidisa. "Domain and task adaptive pretraining for language models." Proceedings http://ceur-ws. org ISSN 1613 (2020): 0073.
> >
> > [I] Devlin, Jacob, et al. "Bert: Pre-training of deep bidirectional transformers for language understanding." arXiv preprint arXiv:1810.04805 (2018).
> >
> > [J] Jinhyuk Lee, Wonjin Yoon, et al. "Biobert: a pre-trained biomedical language representation model for biomedical text mining." Bioinformatics, 36:1234 – 1240, (2019)
> >
> > [K] Ilias Chalkidis, Manos Fergadiotis, et al. "Legal-bert: 'preparing the muppets for court’." ArXiv, abs/2010.02559, (2020).
> >
> > [L] Zixuan Ke, Yijia Shao, et al. "Adapting a language model while preserving its general knowledge." arXiv preprint arXiv:2301.08986, (2023).
> >
> > [M] Zhen Guo and Yining Hua. "Continuous training and fine-tuning for domain-specific language models in medical question answering." ArXiv, abs/2311.00204, (2023).
> >
> > [N] Jiawei Zheng, Hanghai Hong, et al. "Fine-tuning large language models for domain-specific machine translation." ArXiv, abs/2402.15061, (2024).
> >
> > [O] Haoran Yang, Yumeng Zhang, et al. "Unveiling the generalization power of fine-tuned large language models." ArXiv, abs/2403.09162, (2024).

---

### Official Review · Reviewer_4NSY · 2024-07-13

**Soundness:** 4
**Presentation:** 4
**Contribution:** 3
**Rating:** 7
**Confidence:** 4

**Summary:**

The authors propose to add a simple and efficient Cluster-guided Sparse Expert (CSE) layer to Language Models to improve their capability on long-tailed knowledge. The authors demonstrate that pretraining LMs using CSE leads to better performance on domain specific tasks than vanilla pretraining and suggest that the resource intensive finetuning step may not be necessary if the CSE layer can learn long-tailed knowledge properly.

**Strengths:**

1. The paper is well written and concise while providing key insights from a research perspective. The results are also quite strong which demonstrate in all cases using MoA or MoF is better than domain specific finetuning.

2. section 2 is especially quite well written and provides unique perspective into the working of language models. I especially enjoyed reading the gradient consistency analysis which is quite intuitive and easy to understand.

3. I believe if the big players are able to use this method and this works for larger models, it could benefit the GPU poor a lot as they dont have to finetune their models and just need to worry about inference. This method has the potential to ensure LLMs can reach a wider audience.

**Weaknesses:**

Small nitpicks

Typo in line 189: Should be "Cluster" and not "Clsuter"

Typo in Table 1, 4th row: Should be "legal" and not "lgeal"

Its better if the limitations are in the main paper and not in appendix.

**Questions:**

1. I understand that doing this experiment for bigger models on academic resources is a tough ask. But I still ask if the authors have any intuition on what might happen if this method is applied to larger models? My second question is in a larger model, do the authors think the GC for long-tailed knowledge will be even lower?

2. Can this method be applied to both FFN and Attention modules. I am curious since in some cases MoA is better and in some cases MoF is better and was wondering whether using both would lead to the best results.

**Limitations:**

The authors have address limitations but not in the main paper.

---

> ### Author Rebuttal · Authors · 2024-08-06
>
> > Q1 Typo in line 189: Should be "Cluster" and not "Clsuter"; Typo in Table 1, 4th row: Should be "legal" and not "lgeal".
>
> **A1** Thank you for your comments. In our revised version, the error on line 189 will be corrected to _Cluster_, and the typo in Table 1, fourth row, will be updated to _legal_.
>
> > Q2 Its better if the limitations are in the main paper and not in the appendix.
>
> **A2** Thanks for your suggestion. We agree that including the limitations in the main paper rather than the appendix will enhance clarity for readers. We will make this adjustment in our revised document.
>
> > Q3 I understand that doing this experiment for bigger models on academic resources is a tough ask. But I still ask if the authors have any intuition on what might happen if this method is applied to larger models? My second question is in a larger model, do the authors think the GC for long-tailed knowledge will be even lower?
>
> **A3** Thank you for your comments. Our latest experiments, which included 330M GPT-style models and tasks from more domains as detailed in the global rebuttal, demonstrate that our method continues to outperform baseline methods at a larger scale.
> Your question about the gradient consistency (GC) of long-tailed knowledge in larger models is crucial, as it directly relates to how our method would adapt to larger scales. We've identified three key factors influencing this:
>
> - **Model Size** Larger models have greater learning capacities, which can enhance the GC for long-tail knowledge given a fixed training data size.
>
> - **Window Size** A larger window size allows the model to capture
> longer-range relationships within each training sequence, potentially highlighting more subtle distinctions and leading to a more long-tail distribution.
>
> - **Training Data Size** Larger data sizes and larger window sizes might result in a higher proportion of common data relative to long-tail clusters. The increased presence of common data and the potential subdivision of long-tail clusters into smaller groups could, paradoxically, reduce the GC for long-tail knowledge.
>
> > Q4 Can this method be applied to both FFN and Attention modules? I am curious since in some cases MoA is better and in some cases, MoF is better, and was wondering whether using both would lead to the best results.
>
> **A4** Thank you for your suggestion. We tried this architecture and tested it in the same small-scale setting reported in the paper, and found that MoA+MoF yields improvements in the overall performance compared to MoA/MoF only.
>
> | Models | Overruling | Casehold | GAD      | EUADR     | SST2      | average  |
> |--------|------------|----------|----------|-----------|-----------|----------|
> | MoA    | 86.62      | **50.94**| 72.90    | 90.09     | 66.60     | 73.43    |
> | MoF    |  **89.10** | 50.82    | 71.65    | 91.23     | **67.98** | 74.16    |
> | MoA+MoF| 87.50      | 50.83    | **79.87**| **93.80** |  67.09    | **75.82**|

---

### Official Review · Reviewer_YnWw · 2024-07-14

**Soundness:** 3
**Presentation:** 3
**Contribution:** 3
**Rating:** 7
**Confidence:** 3

**Summary:**

This paper proposes a novel approach called Cluster-guided Sparse Experts (CSE) to improve language models' ability to learn long-tail domain knowledge during pretraining, potentially eliminating the need for domain-specific finetuning. This study introduced CSE layers that cluster semantically similar long-tail data and assign it to specialized experts for dedicated learning. Empirical  experiments demonstrate that CSE-based models outperform regularly pre-trained and finetuned models on various downstream tasks without domain-specific finetuning.

**Strengths:**

The CSE method offers an innovative solution to the challenge of learning long-tail domain knowledge in language models. The paper provides an in-depth analysis of gradient behaviors, embedding spaces, and cluster dynamics to support its claims. This approach could significantly reduce the need for costly and time-consuming domain-specific finetuning.

**Weaknesses:**

Typo: line 189/300 should be Cluster rather than Clsuter.

**Questions:**

No

---

> ### Author Rebuttal · Authors · 2024-08-06
>
> We appreciate the time and expertise you have invested in reviewing our submission. Below, we outline our responses to the specific points raised in your reviews.
>
> > Q1 Typo: line 189/300 should be Cluster rather than Clsuter.
>
> **A1** Thank you for your comments. It has been corrected to be _clusters_ in our revised document.

---

### Official Review · Reviewer_1S4s · 2024-07-28

**Soundness:** 3
**Presentation:** 3
**Contribution:** 3
**Rating:** 6
**Confidence:** 4

**Summary:**

The paper presents an innovative approach to address the challenge of finetuning language models (LMs) for domain-specific tasks. The authors argue that the traditional pretraining-finetuning paradigm is suboptimal due to the high cost and time consumption of finetuning. To tackle this, the authors propose the Cluster-guided Sparse Expert (CSE) layer, which is designed to enhance the model's awareness of long-tail knowledge during the pretraining phase, eliminating the need for finetuning.
The CSE layer leverages the intuition that semantically similar data points are closer in the embedding space. It employs an efficient clustering mechanism during pretraining to group long-tail domain data into distinct clusters and assigns additional experts to memorize this knowledge. The paper demonstrates that with this strategy, the language models can achieve superior performance on various downstream tasks compared to traditionally pretrained-finetuned LMs.

**Strengths:**

I'm not that familiar with the moe and pretraining field, but I like the motivation and idea of this method despite the difficulty of verifying the idea in larger models in academic institutions. I agree with the author's claim and the methods proposed a new way for the future moe training.

**Weaknesses:**

1. The improvement in the GPT style model is subtle compared to the baseline moe method, which may undermine the advantage of the proposed methods.
2. The experiments are not sufficient, maybe more tasks should be tested. Is it possible to conduct the experiments on an pre-trained model, which would make the contribution stronger.

**Questions:**

1. In Figure 3(c), the legend is lines 1-12. Maybe the author wants to show layers?
2. Can you conduct computation time and memory analysis of the proposed methods as you claim that fine-tuning is time-costing.

**Limitations:**

Yes

---

> ### Author Rebuttal · Authors · 2024-08-06
>
> We appreciate the time and expertise you have invested in reviewing our submission. Below, we outline our responses to the specific points raised in your reviews.
>
> > Q1 The improvement in the GPT style model is subtle compared to the baseline MoE method, which may undermine the advantage of the proposed methods.
>
> **A1** Thank you for your comments. The experiments presented in the paper are conducted with 130M models, and we've conducted larger-scale experiments with 330M GPT-style models and detailed the results in the global rebuttal. Results indicate that our approaches match and outperform the baseline MoE model, achieving an average accuracy improvement of 3.26%.
>
> > Q2 The experiments are not sufficient, maybe more tasks should be tested. Is it possible to conduct the experiments on a pre-trained model, which would make the contribution stronger?
>
> **A2** Thank you for your comments. To further substantiate the efficacy of our method, we've conducted experiments on a wider array of tasks drawn from diverse domains using the 330M GPT-style model. As detailed in the global rebuttal, we added domain-specific tasks from the academic, environmental, and financial sectors to demonstrate the generalization capability of our method across multiple domains. Additionally, we have included results on general knowledge tasks to confirm that our dedicated long-tail learning method does not compromise general knowledge acquisition.
>
> We also conducted experiments on a pre-trained 110M scale model as you suggested, wherein all methods continue training from a single pre-trained checkpoint. The outcomes are presented in the following table. Results show that directly applying our method to a pre-trained model still yields superior performance compared to the baseline and MoE models.
>
> | Models     | Overruling |  Casehold |    GAD    |   EUADR   |   SST2    |  average  |
> |------------|------------|-----------|-----------|-----------|-----------|-----------|
> | BERT/med   |    85.00   |   51.11   |   70.84   |   89.86   | **64.11** |   72.18   |
> | BERT/legal |    85.83   | **51.21** |   66.08   |   88.45   |   61.58   |   70.63   |
> | MoE/med    |    85.83   |   49.91   |   72.18   |   90.42   |   63.42   |   72.35   |
> | MoE/legal  |  **86.67** |   50.83   |   69.44   |   89.86   |   62.16   |   71.88   |
> | Ours/MoA   |  **86.67** |   51.11   |   73.17   | **92.96** |   63.99   | **73.58** |
> | Ours/MoF   |  **86.67** |   50.77   | **73.25** |   91.83   |   63.65   |   73.23   |
>
> > Q3 In Figure 3(c), the legend is lines 1-12. Maybe the author wants to show layers?
>
> **A3** Thank you for your comments. We'll correct it to "layers 1-12" instead of "lines" in our revised document.
>
> > Q4 Can you conduct computation time and memory analysis of the proposed methods as you claim that fine-tuning is time-costing?
>
> **A4** Our method introduces negligible overhead in time and memory.
>
> - **Memory** The additional memory consumption incurred by our method compared with an MoE who has an equivalent number of experts is attributable to the storage requirements for $k$ distinct cluster centers, each endowed with a dimension of $d'$, which is $O(kd')$ in total. In most practical scenarios, the $k$ is lower than a hundred, and $d'$ tends to be around a hundred. Thus $O(kd')$ is negligible when compared with a neural network with often millions of parameters. Moreover, during a forward pass, our model mirrors both the Mixture of Experts and baseline architectures in the number of parameters activated, thus not introducing a memory burden.
>
> - **Time** Throughout the training phase, the additional computation introduced by our method, relative to both the baseline and MoE, is the cluster initialization step. The time complexity of this operation will be $O(n\log n d')$ and is bounded by $O(n^2d')$ for the worst case, where $n$ represents the number of sampled points. Notably, during subsequent inference stages, the quantum of parameters activated per iteration by our algorithm aligns identically with that of both MoE and the baseline.
>
> Since MoE and the baseline necessitate an additional fine-tuning phase, it becomes evident that the time cost of this phase far surpasses the $O(n^2d')$ complexity of cluster initialization. To provide more straightforward data, we list the wall-clock time of training & finetuning 330M GPT on 20B tokens with 2 NVIDIA GeForce RTX 3090 GPUs each in the table below.
>
> | Model       | Baseline GPT   | MoE   | Ours |
> |-------------|---------------|-------|------|
> | Time(hrs)   | 177            | 205   | 160  |

---

> > ### Comment · Reviewer_1S4s · 2024-08-10
> >
> > Thank you for your responses.
> >
> > The authors have addressed some issues and I'm glad to see the method's effectiveness on the pre-trained model. I think if the work can be extended to a larger model efficiently, it would be better.
> > For example, it would be exciting if we could train an extra moe for LLAMA to a domain-specific model. Is it possible for the proposed method?
> >
> > I struggle with what score should I give as I think it is a good point but the scenario is mainly training a new model in a small scale.
> > I currently decided to keep my score and follow-up as additional reviewer comments come in.

---

> > > ### Author Response · Authors · 2024-08-14
> > >
> > > > DQ1 Thank you for your responses. The authors have addressed some issues and I'm glad to see the method's effectiveness on the pre-trained model. I think if the work can be extended to a larger model efficiently, it would be better. For example, it would be exciting if we could train an extra moe for LLAMA to a domain-specific model. Is it possible for the proposed method?
> > >
> > > > I struggle with what score should I give as I think it is a good point but the scenario is mainly training a new model in a small scale. I currently decided to keep my score and follow-up as additional reviewer comments come in.
> > >
> > >
> > > __DA1__ Thank you for your comments. The idea you proposed of applying our methodology to pre-trained LLM is indeed valuable. We are also interested in it and are looking forward to testing our approach on larger models.
> > >
> > > Our existing experiments have shown that, as the model scales from 130 million to 330 million parameters, the patterns within the model's representational space, along with the performance, remain consistent. This consistency lends us confidence that our methodology will continue to be effective when applied to even larger models.
> > >
> > > Consequently, we have developed an experimental plan to test our methodology by fine-tuning a 7B LLaMA model. We have collected a diverse set of pre-training and downstream task datasets from multiple domains, including legal (CaseLaw, CONTRACTS), biomedical (PMC Full-text articles, BC5CDR, BC4CHEMD), finance (TRC2, ConvFinQA), code (Stack, codex), and NLP academic literature (ACL-ARC, EBM-NLP), among others. We have subscribed commercial computational resources, and we estimate that completing these experiments will require approximately 35 days.
> > >
> > > Upon completing these experiments, we will incorporate the findings into our revised article. Should the paper be accepted for publication, we will include these results in the camera-ready version.

---

### Official Review · Reviewer_bZcS · 2024-07-29

**Soundness:** 3
**Presentation:** 3
**Contribution:** 3
**Rating:** 6
**Confidence:** 4

**Summary:**

The paper focuses on the problem of modeling long-tail domain knowledge in language models, which can be missed during the general-purpose pretraining. While most approaches capture long-tail domains as a second domain-specific pretraining step, the paper proposes a cluster-guided method that encourages the model to learn long-tail domain knowledge during the initial pretraining stage. Specifically, the paper first conducts analysis to show the challenges in learning long-tail domain data and then uses these findings to propose a cluster-guided Sparse Expert (CSE) architecture, which routes sequences from long-tail clusters into separate expert modules (either attention or FFN modules of the last model layer). The proposed method is evaluated on a simulated setup with common domains drawn from wikipedia data and long-tail domains drawn from legal and medical datasets.

**Strengths:**

The paper is clearly written and easy to follow. The authors did a great job presenting a coherent analysis to motivates the design choices behind their proposed CSE method. CSE is promising as it alleviates the need to resort into a second training stage, which would require careful data-replay or regularization. CSE is also empirically shown to be effective for modeling legal and medical domain knowledge.

**Weaknesses:**

The experimental results raise questions about the effectiveness of CSE and baselines.
1. The baseline results in Tables 1 and 2 are surprising as, for example, BERT/med (further trained on medical) would be expected to outperform BERT/legal (further trained on legal) on medical datasets, but the two models are instead performing very similarly to each other, which suggests potential issues with baseline tuning.
1. It is not clear if the proposed method retains its general knowledge as there are not sufficient benchmarks to judge general knowledge (e.g., WebQS, TriviaQA).

The question "is finetuning indispensable to LMs?" and argument that "Finetuning improves LMs’ domain performance by providing a second lesson, which could be avoided if the first (pretraining) is appropriately delivered." could be misleading as finetuning often serves multiple roles, including instruction tuning (as in SFT). To avoid misunderstanding, I would suggest being more precise, e.g., by changing "finetuning" into "domain-specific pre-training".

**Questions:**

Why is k-means clustering used for the analysis but DBSCAN is used for the main method? What is the value of K in k-means?

Can the authors clarify the reasoning behind routing at sequence-level instead of token-level?

Are there any recommendations for adapting to more frequent (instead of long tail) domains?

**Limitations:**

Yes

---

> ### Author Rebuttal · Authors · 2024-08-06
>
> We appreciate the time and expertise you have invested in reviewing our submission. Below, we outline our responses to the specific points raised in your reviews.
>
> > Q1 The baseline results in Tables 1 and 2 are surprising as, for example, BERT/med (further trained on medical) would be expected to outperform BERT/legal (further trained on legal) on medical datasets, but the two models are instead performing very similarly to each other, which suggests potential issues with baseline tuning.
>
> **A1** Thank you for your comment. Upon investigation, we discovered that the issue stemmed from catastrophic forgetting during the fine-tuning stage. Our training process strictly followed the basic framework of pretraining and domain-specific pretraining, and we didn't control the occurrence of forgetting, focusing more on the best performance. However, we monitored the forgetting phenomenon using the test loss on pretraining data. Specifically, BERT/med(further trained in medical), exhibited a more severe forgetting issue, with test perplexity on pretraining data increasing by 24.33, compared to a smaller increase of 8.41 for BERT/legal(further trained in legal). After correcting this imbalance by utilizing early-stop to select checkpoints where each model showed similar degrees of forgetting but not the best performance, the results came out that each domain-finetuned model outperformed on its respective domain tasks. This further underscores the challenges posed by the fine-tuning stage, affirming the value of our approach in not requiring domain-specific pretraining.
>
> | Models     | Casehold | Overruling | GAD | EUADR |
> |------------|-----------|------------|-----|-------|
> | BERT/legal | **50.26** | **85.83** | 62.01 | 79.15 |
> | BERT/med   | 49.15     | 85.00     | **64.58** | **81.97** |
>
> > Q2 It is not clear if the proposed method retains its general knowledge as there are not sufficient benchmarks to judge general knowledge (e.g., WebQS, TriviaQA).
>
> **A2** Thank you for your comment. To address this, we've conducted experiments on general knowledge tasks, the results of which are detailed in the global rebuttal. These results demonstrate that our methods not only match but sometimes even surpass the performance of baseline models on a variety of general tasks. This is because our method enhances the model's capability to learn long-tail data by segregating the learning process between long-tail and non-long-tail data, allowing each category to be learned effectively without compromising one at the expense of improving the other.
>
> > Q3 Why is k-means clustering used for the analysis but DBSCAN is used for the main method? What is the value of K in k-means?
>
> **A3** Thank you for your comments. Initially, we used the K-means clustering algorithm with the elbow method to identify the optimal number of clusters (K). However, the elbow method's reliance on subjective judgment is a limitation. We then tested DBSCAN, which automatically determines the number of clusters based on density and is easier to set parameters compared to K-means. Consequently, we opted to employ DBSCAN in our solution design. In the global rebuttal Figure 1, we also presented cluster number and cluster distance calculated with DBSCAN, demonstrating that the choice of clustering algorithm does not impact the statements presented in Section 2.
>
> > Q4 Can the authors clarify the reasoning behind routing at sequence-level instead of token-level?
>
> **A4** Thank you for your comments. The definitions, analyses, and conclusions regarding long-tail data presented in Section 2 are all at the sequence level; consequently, routing is also performed at this same level. The rationale for defining and analyzing long-tail at the sequence level stems from the fact that when we discuss long-tails, we are essentially referring to semantics rather than individual words or tokens. Such semantics can only be observed within the context provided by sequences as a whole.
>
> > Q5 Are there any recommendations for adapting to more frequent (instead of long tail) domains?
>
> **A5** Thank you for your question. As discussed above, since data from frequent domains dominate the gradient direction of model optimization, our method separates the parameter of learning long-tail and frequent domains to improve the model's ability to capture long-tail domain knowledge. If we expect to improve the ability on frequent domains, we may need to design methods to analyze the gradient consistency within frequent clusters, and conduct for finer distinctions within the frequent domain.

---

### Author Rebuttal · Authors · 2024-08-06

# Global Rebuttal

Dear AC and Reviewers,

We sincerely appreciate the time and expertise you devoted to reviewing our submission.

Given that experiments involving larger scales or a greater number of tasks is a shared concern among the reviewers, we have detailed the outcomes of such experiments in the global rebuttal.

As listed below, we used larger models (330M GPT-style models trained with 20B tokens) and added more domain-specific tasks from the academic, environmental, and financial domains to demonstrate the generalization capability of our method. Additionally, we provided results on general knowledge tasks to show that our method does not impact general knowledge acquisition. The results indicate that our method, even without fine-tuning, consistently outperforms baselines by an average of 3-5\% in accuracy on domain-specific tasks while maintaining comparable performance on general tasks. This highlights the effectiveness of our approach.

In the pdf file, Figure 1 is for Reviewer bZcS and Figures 2,3,4 are for reviewer P2qX.

---
**Results of new tasks tested on GPT 330M trained with 20B tokens**

| Task           | Domain       | Freq. Score | Baseline(tuned) | MoE(tuned) | Ours(w/o tune)  |
|----------------|--------------|-------------|-----------------|------------|-----------------|
| chem-prot      | academic     | 0.207       | **36.25**       | **36.25**  | **36.25**       |
| MAG            | academic     | 0.324       | 63.22           | 64.91      | **65.47**       |
| rct-20k        | academic     | 0.261       | 76.95           | 78.28      | **80.15**       |
| climate detect | environment  | 0.376       | 78.94           | 79.90      | **80.26**       |
| climate sent   | environment  | 0.317       | 66.81           | 68.31      | **69.98**       |
| FPB            | financial    | 0.243       | 16.83           | 25.00      | **40.11**       |
| **average**    |**domain-spec**| -          | 56.50(-5.54)    | 58.77(-3.26) | **62.04**     |
| COLA           | general      | 0.389       | 69.10           | 69.10      | **69.20**       |
| QNLI           | general      | 0.325       | **60.17**       | 60.06      | 59.72           |
| MRPC           | general      | 0.343       | 70.18           | 71.75      | **71.98**       |
| QQP            | general      | 0.380       | 73.28           | 74.47      | **75.95**       |
| SST2           | general      | 0.327       | 74.50           | 72.03      | **76.00**       |
| **average**    | **general**  | -           | 69.45(-1.12)    | 69.48(-1.09) | **70.57**     |

---
**Results of general tasks on BERT with the same small-scale setting in the paper**

| Task        | Domain   | Freq. Score | Baseline(tuned) | MoE(tuned) | Ours(w/o tune) |
|-------------|----------|-------------|-----------------|------------|-----------------|
| COLA        | general  | 0.389       | 69.22           | 69.32      | **69.42**       |
| QNLI        | general  | 0.325       | 65.92           | 65.07      | **66.04**       |
| MRPC        | general  | 0.343       | **72.06**       | 70.83      | **72.06**       |
| QQP         | general  | 0.380       | **70.64**       | 69.85      | 70.18           |
| SST2        | general  | 0.327       | 66.86           | 64.79      | **67.98**       |
| **average** | general  | -           | 68.94(-0.20)    | 67.97(-1.26) | **69.14**       |

---

### Decision · Program_Chairs · 2024-09-25

**Decision:**

Accept (poster)

**Comment:**

The paper introduces a novel Cluster-guided Sparse Expert (CSE) layer for language models that improves the capture of long-tail domain knowledge during pretraining, potentially reducing the need for domain-specific fine-tuning.  The idea of CSE layers is novel and the effectiveness of the idea in learning long-tail knowledge has been demostrated in the experiments.  In-depth analysis has been conducted.  Despite some concerns about presentation quality, the overall clarity of the paper and the coherence of the analysis are praised, making the paper's contributions understandable to the reader.  The authors address the reviewers' concerns in details in the rebuttal.